# Short communication: Timber harvesting impacts small mammal foraging behavior and larval tick infestation

Stephanie N. Hurd [1,2]*, Allison M. Gardner[1]

1 School of Biology and Ecology, University of Maine, Orono, Maine, United States of America,
2 Department of Biology, University of Massachusetts Boston, Boston, Massachusetts, United States of America

* stephanie.hurd@maine.edu

## Abstract

Small mammals are important blood-meal hosts for the blacklegged tick, *Ixodes scapularis*, and reservoirs for the pathogens it transmits. Timber harvesting, a widespread forest management practice within *I. scapularis*'s endemic range, may impact tick densities and infection prevalence via effects on rodent communities. We compared rodent population size, activity patterns, tick burdens, and tick-borne pathogen infection rates in forests under different timber harvesting regimes. We found that harvest intensity correlates negatively with rodent foraging intensity and positively with tick burdens. Thus, host behavior may mechanistically link forest structure to tick densities in managed forests.

## Introduction

Small mammals play a significant role in tick-borne pathogen transmission, serving as both pathogen reservoirs and blood-meal hosts for ticks. Throughout much of North America, the white-footed mouse (*Peromyscus leucopus*), deer mouse (*Peromyscus maniculatus*), and eastern chipmunk (*Tamias striatus*) serve as key hosts for immature blacklegged ticks (*Ixodes scapularis*) [1,2]. *Ixodes scapularis* is the vector for *Borrelia burgdorferi*, *Anaplasma phagocytophilum*, and *Babesia microti*, the causative agents of Lyme disease, human granulocytic anaplasmosis, and human babesiosis, respectively. Rodent foraging behavior may alter tick-host encounter rates, which determine tick burdens (i.e., the number of ticks present on an individual host). In turn, larger tick burdens increase pathogen amplification in both tick and small mammal communities and transmission risk in forest ecosystems [3].

Because rodent behavior responds to habitat configuration [4], common forest management practices that determine forest structure, like timber harvesting, may influence interactions between ticks and their small mammal hosts. Despite previous studies finding local reductions in tick densities [5–8] and small mammal capture

**Data availability statement:** All .csv files are available from the Mendeley Data database (DOI:10.17632/stwjg4ddym.1).

**Funding:** This study was supported by a USDA National Institute of Food and Agriculture grant (Project No. ME012450318; https://www.nifa. usda.gov/) to AMG and Hatch award through the Maine Agricultural and Forest Experiment Station (Project No. ME032025; https://umaine. edu/mafes/) to AMG. No funders played a role in the study design, data collection or analysis, decision to publish, or manuscript preparation.

**Competing interests:** The authors have declared that no competing interests exist.

rates following a recent timber harvest [5,6], no field studies have empirically linked timber harvesting, small mammal behavior, and ticks. Our previous study examined the effect of timber harvesting on forest structure, microclimate, deer, and ticks, but lacked any investigation of small mammal communities [8]. Timber harvesting also alters small mammal habitat and predation pressure through changes in overstory and understory vegetation cover, potentially affecting small mammal population size [9] or activity. Both small mammal population size and foraging behavior may affect availability of hosts to ticks.

No research to date has investigated impacts of timber harvesting on foraging behavior of *I. scapularis* hosts and the potential consequences for infected tick densities. Moreover, while timber harvesting alters forest overstory structure, even in managed stands microhabitat conditions on the forest floor vary, which could influence small mammal sensitivity to predation risk [10]. Thus, this study's objectives were 1) to determine correlations between timber harvesting intensity and small mammal population size, overall activity (measured as the abundance of animal tracks recorded on track plates), foraging behavior, mean tick burden, and pathogen infection prevalence, and 2) to assess the importance of understory vegetation cover on small mammal foraging behavior within stands with different timber harvesting histories.

## Methods

Our study took place from 9–27 August 2021. We conducted a spatially replicated experiment in a blocked design. Three towns (New Gloucester, Lyman, and Standish) in southern Maine, USA, with high incidence of *I. scapularis* and tick-borne illness in humans [11,12] were the blocks, each containing two replicate forest stands spaced at least 500 m apart to maintain independence, for a total of six experimental units. Each experimental unit was a 1-ha square within an eastern white pine – mixed hardwood forest stand [13]. Previous work documented daily deer activity was consistent across these stands during the two years preceding this study [8]. Specifically, detections ranged from 0.1 to 0.3 deer per day, with an Analysis of Variance (ANOVA) reporting no significant variation between these six experimental units ($F=3.4$, $p=0.1$). Dominant tree species present in all stands included red oak (*Quercus rubra*), American beech (*Fagus grandifolia*), sugar maple (*Acer saccharum*), red maple (*Acer rubrum*), eastern white pine (*Pinus strobus*), and eastern hemlock (*Tsuga canadensis*). Each replicate stand represented one of two treatments of different partial timber harvesting intensities. High harvest intensity stands had trees harvested in the past 10–20 years while low harvest intensity stands had little to no tree removal. High harvest stands had, on average, fewer trees per hectare (129.0–378.6 trees) and lower basal area (i.e., area occupied by tree stems, 5.0–17.5 m2) per hectare, but higher sapling counts (5.6–10.0 saplings) and understory vegetation cover (45–75%). Comparatively, low harvest stands had more trees per hectare (546.9–669.2 trees) and greater basal area per hectare (26.6–41.3 m2), but lower sapling counts (0.4–0.6 saplings) and understory vegetation cover (21–57%).

Although most tick-borne disease studies use live trapping to characterize small mammal communities, we used an additional two techniques: foraging trays and track plates. We used these techniques in tandem because each provides information about different small mammal community characteristics. We collected data on a 50 m × 100 m grid spaced at 10 m intervals placed in the center of the hectare to minimize edge effects. In each experimental unit, every night for three consecutive trap nights, we deployed 50 H.B. Sherman live traps (catalog #LFA, H.B. Sherman Traps, Inc., Tallahassee, FL; University of Maine IACUC protocol #A2018-11–06), 12 foraging trays (six placed beneath understory cover and six in the open) to calculate giving-up densities (GUDs) [14], and 16 track plates.

We conducted small mammal trapping and handling in full accordance with the regulations and guidelines provided by the written consent of the University of Maine IACUC protocol #A2018-11–06. During live trapping, we applied a numbered ear tag to each animal (catalog #1005−1, National Band and Tag Company, Newport, KY), identified the animal to species, and collected an ear biopsy. To test for pathogen infection and differentiate between *P. leucopus* and *P. maniculatus*, a multiplex quantitative polymerase chain reaction (qPCR) assay was performed as described in Rounsville et al. (2021) [12], with additional positive controls to reduce errors in *Peromyscus* species identification, consisting of voucher tissue samples from the University of New Hampshire mammal collection.

We arranged foraging trays to create four distinct conditions depending on the forest stands' harvest intensity and, within each stand, whether the trays were covered or uncovered by understory vegetation: 1) low intensity harvest, covered trays, 2) low intensity harvest, uncovered trays, 3) high intensity harvest, covered trays, and 4) high intensity harvest, uncovered trays. This placement allowed us to evaluate whether overstory changes due to management or microhabitats that exist within both managed and unmanaged areas impact foraging. Each foraging tray was filled with a sand and seed mixture consisting of 1.25 L of fine-grain terrarium sand and 3.00 g of white millet seed, a commonly used bait for small mammals [14–16]. Throughout the sampling period, we returned to the foraging tray every 24 hrs to sift out remaining seed using a fine mesh strainer, which we collected for future weighing, stored in plastic bags. We then "recharged" the foraging tray with 3.00 g of new seed. To calculate the percentage of seed consumed within each foraging tray per sampling night, the collected seed was weighed on a kitchen scale (Taylor Digital Compact Food Scale, Item #070-05-4846). Foraging trays were made from 38.1 cm x 20.3 cm x 12.7 cm plastic containers, with a 5.7 cm diameter hole cut into each end. All trays had plastic covers to ensure no larger mammals, like deer, could remove seed and to prevent any rainfall from washing away sand or seed or changing the sand's consistency. We positioned one trail camera at each sampling station to have a full view of the foraging trays to verify that only small mammals removed seeds.

Track plates were constructed by painting a graphite mixture of approximately 30 mL graphite powder, 5 mL mineral oil, and 180 mL ethanol [17] onto an acetate sheet, which we fastened to aluminum flashing backing with paper clips. We set these track plates with a few sunflower seeds for bait and then collected them 24 hrs later, for three consecutive sampling days. To quantify the percent surface area occupied by track plates, a picture of each track plate was uploaded to two image processing software; first to GNU Image Manipulation Program (GIMP) [18] then ImageJ [19]. In GIMP, the image was cropped to only include the surface area of the track plate. Using ImageJ, we improved the image contrast to distinguish between animal tracks and the rest of the undisturbed track plate, converted the picture to a binary black and white image, and then analyzed the percentage of pixels containing animal tracks.

Data analyses were conducted using paired t-tests and ANOVA in R version 4.0.2 (R Foundation for Statistical Computing, Vienna, Austria).

## Results

We live-captured 158 small mammals across low harvest stands and 96 small mammals across high harvest stands. All mice were *P. leucopus*. The most abundant species across low harvest stands were *P. leucopus* (n = 153), *T. striatus* (n = 4), and *Clethrionomys gapperi* (n = 1). Across high harvest stands, the most abundant

species were *P. leucopus* (n = 79), *M. gapperi* (n = 10), *T. striatus* (n = 6), and *Napaeozapus insignus* (n = 1). Due to the dominance of *P. leucopus* and very low capture frequencies of all other small mammals, only *Peromyscus spp.* capture data were analyzed. Tick burdens consisted of only larval ticks, except for a single adult and a few nymphs, so these latter life stages were excluded from analysis. *Peromyscus leucopus* in the high harvest treatment had higher larval tick burdens than those in the low harvest treatment ($F = 9.90$, $df = 1, 220$, $p < 0.01$, Fig 1a). However, small mammals foraged more and consumed more seed in the low harvest treatment ($t = 2.85$, $df = 107$, $p < 0.01$, Fig 1b). In low harvest stands, small mammals foraged more and consumed more seed in foraging trays covered by understory vegetation than in uncovered trays ($t = 2.61$, $df = 53$, $p = 0.012$, Fig 1c), but in high harvest stands, seed consumption did not differ between covered versus uncovered foraging trays ($t = 1.52$, $df = 53$, $p = 0.13$, Fig 1c). These results indicate the importance of understory conditions in managed forest stands, consistent with our previous work emphasizing the influence of microhabitat features within the Lyme disease system [8]. This study is the first to simultaneously document evidence of timber harvesting influences on small mammal foraging behavior and mean larval tick burden.

We detected three tick-borne pathogens in *P. leucopus*. The highest proportion were positive for *B. burgdorferi* (29.52%), followed by *A. phagocytophilum* (7.14%), and *B. microti* (4.76%). There was no significant effect of harvesting

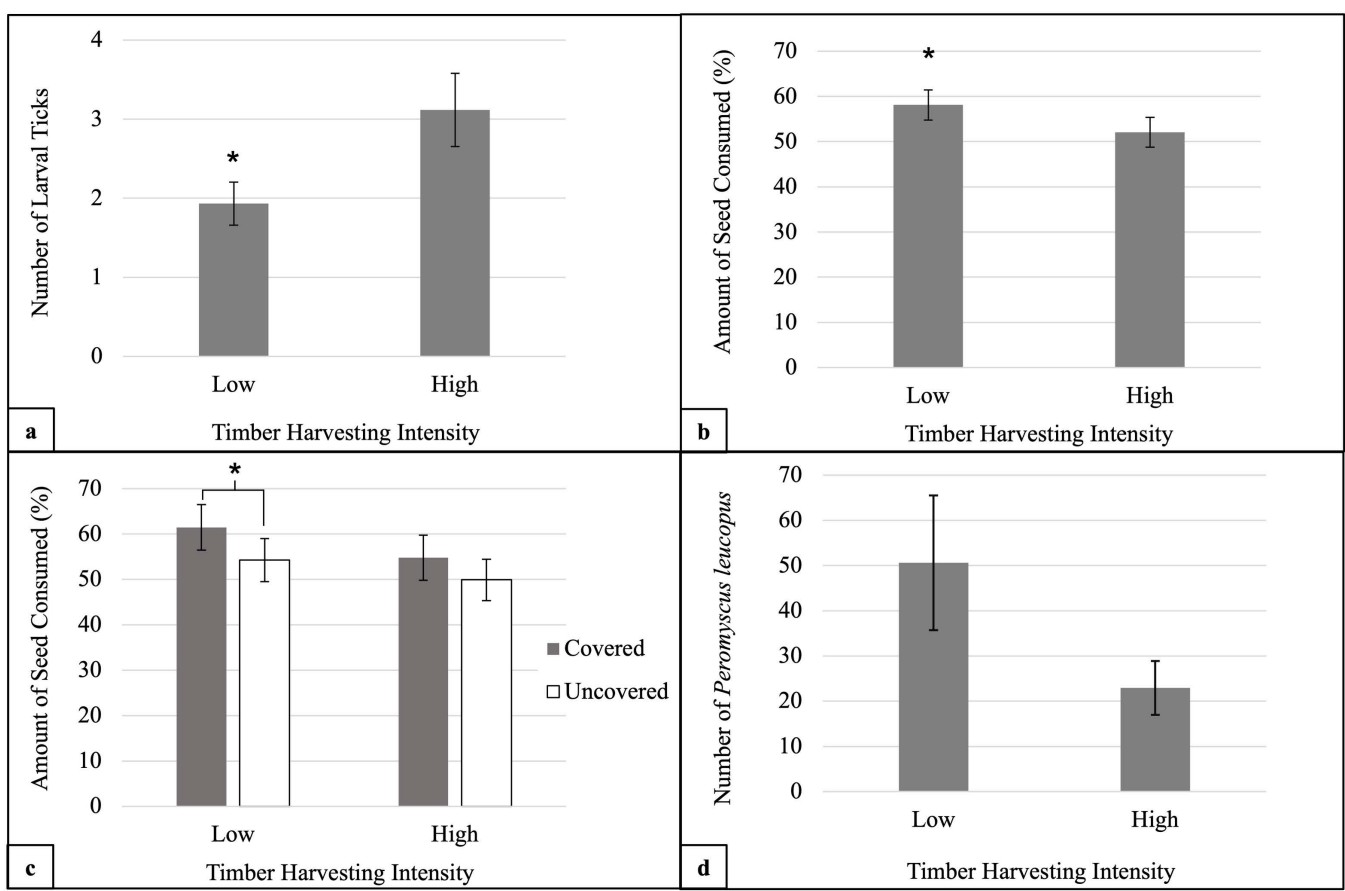

**Fig 1. Timber Harvesting Intensity and Small Mammal Population Dynamics.** Timber harvesting treatment versus small mammal a) average larval tick burden, b) average percent seed consumed, c) average percent seed consumed by foraging tray coverage, and d) population size estimates. Asterisk denotes statistical significance at α = 0.05.

treatment on tick infection prevalence for any pathogen. The amount of small mammal tracks on the track plates did not differ between harvest treatments ($F = 0.23$, $df = 1, 239$, $p = 0.63$). Nor did *P. leucopus* population size ($t = 3.10$, $df = 2$, $p = 0.09$, Fig 1d), although, this may reflect a high estimated Type II error rate ($\beta = 0.71$) due to our small number of stands ($n = 3$ pairs); additional research should further explore rodent population size response to timber harvesting, as past work showed reduced small mammal survival [20] and capture rates [5] post-harvest.

We found that small mammal *B. burgdorferi* infection prevalence correlated with population size estimates ($F = 76.52$, $df = 1, 4$, $p = 0.013$) and GUDs ($F = 25.26$, $df = 1, 4$, $p = 0.037$). Population size and GUDs did not correlate with small mammal tick burden, *A. phagocytophilum* infection prevalence, or *B. microti* infection prevalence. Track plate activity did not correlate with tick burden or infection prevalence.

## Discussion

This report is the first comparison of these sampling methods' abilities to predict small mammal tick burdens and infection prevalence, demonstrating the potential utility of these techniques. While live trapping is widely used and provides detailed information on individual animals, it is costly, time- and labor-intensive, and invasive. Past studies in tick-borne disease research have used GUDs to investigate white-tailed deer (*Odocoileus virginianus*) foraging response to lone star tick (*Amblyomma americanum*) parasitism risk [21], but to our knowledge this is the first use of this method in measuring rodent behavioral response to timber harvesting, specifically. We show that foraging trays could be a complementary or alternative method to live trapping in predicting tick-borne pathogen infection prevalence, with the added benefits of providing foraging behavior information and requiring less sampling effort.

A limitation of track plates and foraging trays is the inability to determine which small mammal species were present. Trail cameras deployed with the track plates and foraging trays verified that only small mammals visited, but photos were not detailed enough to distinguish species. In the Lyme disease system, small mammals vary in reservoir competence and permissiveness [1,2]. For example, mice and shrews are highly competent *B. burgdorferi* reservoirs whereas gray (*Sciurus carolinensis*) and red (*Tamiasciurus hudsonicus*) squirrels may have high tick burdens but are less competent reservoirs [1] and often kill ticks through efficient grooming [2]. These species' activity and foraging may have varied consequences for *B. burgdorferi* transmission. Additionally, while foraging trays assess one aspect of small mammal behavior, others not captured using these trays, such as nesting behavior, could prove influential in maintaining the host-pathogen cycle. A study in Northern Wisconsin, USA found *I. scapularis* larvae in 64% of *Peromyscus* spp. nests, and 87% of these ticks were blood-fed [22]. Nesting behavior may influence the abundance of ticks inhabiting nests and, subsequently, opportunities for ticks that detach and molt in the nest to later parasitize highly competent reservoir hosts, ultimately maintaining tick densities and infection prevalence.

Although prior work found limited variation in deer activity across these experimental units [8], as these units were only 1-ha in size, this study was conducted on a spatial scale much smaller than the typical home range for deer [23]. Therefore, this study design may have failed to capture any differences in deer activity. As deer are important reproductive hosts for ticks, their activity can determine the location and abundance of fecund female ticks that drop off and lay eggs, thus influencing local larval tick density [24]. Consequently, deer activity may influence small mammals' larval tick burden, independent of timber harvesting treatment. This potential confounding variable should be disentangled in future studies of timber harvesting and small mammal tick burden.

The same high harvest intensity forest stands that had lower small mammal GUDs have also been documented to have lower *I. scapularis* densities [8], consistent with previous work that found harvested forest stands had reduced capture rates of small mammals and questing tick presence [5,6]. Thus, our results indicate altered host communities may be a mechanism driving decreased tick densities, possibly in tandem with microhabitat mechanisms as high harvest intensity forest stands had leaf litter with less stable temperatures and lower humidity, which could increase tick mortality [8].

## Acknowledgments

We thank B. Stevens for his assistance with field data collection, the University of Maine Cooperative Extension Tick Lab for performing tick-borne pathogen detection in small mammal tissue samples, and members of the Gardner Lab at the University of Maine for their helpful feedback and support. We thank the U.S. Forest Service Northern Research Station, Maine Woodland Owners, Portland Water District, and individual property owners for land use permission.

## Author contributions

**Conceptualization:** Stephanie N. Hurd, Allison M. Gardner.

**Data curation:** Stephanie N. Hurd.

**Formal analysis:** Stephanie N. Hurd.

**Funding acquisition:** Allison M. Gardner.

**Investigation:** Stephanie N. Hurd.

**Methodology:** Stephanie N. Hurd, Allison M. Gardner.

**Project administration:** Stephanie N. Hurd, Allison M. Gardner.

**Resources:** Allison M. Gardner.

**Supervision:** Stephanie N. Hurd, Allison M. Gardner.

**Validation:** Stephanie N. Hurd.

**Visualization:** Stephanie N. Hurd.

**Writing – original draft:** Stephanie N. Hurd.

**Writing – review & editing:** Stephanie N. Hurd, Allison M. Gardner.

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
