## [Decision Letter · Decision Letter 0]

13 Nov 2024

PONE-D-24-45269Short Communication: Timber harvesting impacts small mammal foraging behavior and tick infestationPLOS ONE

Dear Dr. Hurd,

Thank you for submitting your manuscript to PLOS ONE. After careful consideration, we feel that it has merit but does not fully meet PLOS ONE’s publication criteria as it currently stands. Therefore, we invite you to submit a revised version of the manuscript that addresses the points raised during the review process. The single reviewer has raised issues that require a major revision for resubmission.  In particular, details are required about the study sites and about the developmental stage of the ticks that were the subject of the analysis.  My own comments center on the role of deer as potential confounders to the analysis; there is no mention of these hosts as a critical factor in the abundance and distribution of deer ticks across habitats at any scale.

We look forward to receiving your revised manuscript.

Kind regards,

Sam R. Telford III

Academic Editor

PLOS ONE

“We thank B. Stevens for his assistance with field data collection, the University of Maine Cooperative Extension Tick Lab for performing tick-borne pathogen detection in small mammal tissue samples, and members of the Gardner Lab at the University of Maine for their helpful feedback and support. We thank the U.S. Forest Service Northern Research Station, Maine Woodland Owners, Portland Water District, and individual property owners for land use permission. This study was supported by a USDA National Institute of Food and Agriculture grant (Project No. ME012450318) to AMG and Hatch award through the Maine Agricultural and Forest Experiment Station (Project No. ME032025) to AMG.”

“This study was supported by a USDA National Institute of Food and Agriculture grant (Project No. ME012450318; https://www.nifa.usda.gov/) to AMG and Hatch award through the Maine Agricultural and Forest Experiment Station (Project No. ME032025; https://umaine.edu/mafes/) to AMG. No funders played a role in the study design, data collection or analysis, decision to publish, or manuscript preparation.”

Additional Editor Comments:

I agree with the reviewer's comments; although only one reviewer responded, in the interest of providing you with a timely review I have read the ms myself. I completely agree with the reviewer that the developmental stage of the tick needs to be specified. Indeed, if the trapping was done in August, it is likely that the ticks that are the focus of the analysis are larval deer ticks. This begs the question as to whether differential habitat use by deer has influenced the distribution and abundance of ticks across the two sites. Deer would drop engorged female ticks where they browse the most; such ticks would produce larvae that would infest small mammals in the site. If the ticks infesting small mammals were nymphs, they might have been acquired as larvae elsewhere and transported across sites to drop and develop into nymphs. There needs to be discussion of the potential confounding role of deer in explaining the results that are reported. There also needs to be discussion as to why previously published data requires another analysis and publication.

Reviewers' comments:

Reviewer's Responses to Questions

**Comments to the Author**

1. Is the manuscript technically sound, and do the data support the conclusions?

Reviewer #1: Partly

2. Has the statistical analysis been performed appropriately and rigorously? 

Reviewer #1: Yes

3. Have the authors made all data underlying the findings in their manuscript fully available?

Reviewer #1: Yes

4. Is the manuscript presented in an intelligible fashion and written in standard English?

Reviewer #1: Yes

5. Review Comments to the Author

Reviewer #1: Please see attached document for review comments. More data should be added to make this manuscript as valuable as possible. Given other cited material, the information should be available to the authors.

6. PLOS authors have the option to publish the peer review history of their article (what does this mean? ). If published, this will include your full peer review and any attached files.

**Do you want your identity to be public for this peer review?** For information about this choice, including consent withdrawal, please see our Privacy Policy .

Reviewer #1: No

---

## [Author Response · Author response to Decision Letter 1]

10 Feb 2025

To the Editor:

We appreciate the response provided by PLOS One and the opportunity to revise and resubmit our manuscript and incorporate the thoughtful feedback provided by yourself and the reviewer. Below we provide in-depth responses to the concerns raised and indicate changes made to our manuscript in response to these comments. The line numbers mentioned in our replies to reviewer comments refer to the lines in the document containing tracked changes.

In response to the primary concerns, we have 1) explained the choice to include only larval ticks in our analyses, 2) discussed deer habitat use, 3) described the distinction between this current study and our previous one, and 4) elaborated on our methods and study site descriptions.

Thank you for your consideration and we look forward to your feedback.

Sincerely,

Stephanie Hurd and Allison Gardner

Response to Editor Comments

>>>We have made the necessary changes to ensure our manuscript meets all style requirements, including revising the title page, formatting the entire document, fixing our references, and renaming files.

“We thank B. Stevens for his assistance with field data collection, the University of Maine Cooperative Extension Tick Lab for performing tick-borne pathogen detection in small mammal tissue samples, and members of the Gardner Lab at the University of Maine for their helpful feedback and support. We thank the U.S. Forest Service Northern Research Station, Maine Woodland Owners, Portland Water District, and individual property owners for land use permission. This study was supported by a USDA National Institute of Food and Agriculture grant (Project No. ME012450318) to AMG and Hatch award through the Maine Agricultural and Forest Experiment Station (Project No. ME032025) to AMG.”

“This study was supported by a USDA National Institute of Food and Agriculture grant (Project No. ME012450318; https://www.nifa.usda.gov/) to AMG and Hatch award through the Maine Agricultural and Forest Experiment Station (Project No. ME032025; https://umaine.edu/mafes/) to AMG. No funders played a role in the study design, data collection or analysis, decision to publish, or manuscript preparation.”

>>>We have revised our Acknowledgments section to remove the function information (Lines 296-302). The Funding Statement above contains all necessary information and does not need to be updated.

>>>We have included our full ethics statement in the Methods section (Lines 160-162).

Additional Editor Comments:

I agree with the reviewer's comments; although only one reviewer responded, in the interest of providing you with a timely review I have read the ms myself. I completely agree with the reviewer that the developmental stage of the tick needs to be specified. Indeed, if the trapping was done in August, it is likely that the ticks that are the focus of the analysis are larval deer ticks. This begs the question as to whether differential habitat use by deer has influenced the distribution and abundance of ticks across the two sites. Deer would drop engorged female ticks where they browse the most; such ticks would produce larvae that would infest small mammals in the site. If the ticks infesting small mammals were nymphs, they might have been acquired as larvae elsewhere and transported across sites to drop and develop into nymphs. There needs to be discussion of the potential confounding role of deer in explaining the results that are reported. There also needs to be discussion as to why previously published data requires another analysis and publication.

>>>Thank you for this comment. All the focus of the analysis were larval ticks. We agree it is necessary to make this clear and have clarified the results (Lines 209-210) and changed the manuscript title.

We also agree it is important to address the role of deer in this system, and explain deer are an unlikely confounding variable as past work showed deer habitat use was constant across all sites (Lines 133-133).

While this study uses the same experimental units referred to in our previous publication, the small mammal work was a separate study conducted on only a subset of the total original experimental units (n = 14) after the completion of the prior study in 2020. We attempt to make clear this separation and clarify that we reference the previous study to help speculate on the underlying mechanisms that may explain the findings of this separate project (Lines 100-102).

Response to Reviewer Comments

General comments

One question that occurs to me is why this study, collected during one month in 2021, merits a

stand alone manuscript, when this is obviously part of a larger study that was recently published

and cited in the text (Hurd et al 2024, J Med Ent). Based on the cited study, the overarching project was a two-year endeavor, is there not mammal data for this entire project period?

>>>Thank you for this comment. No, there is no mammal data for the project that occurred from 2019 – 2020. This small mammal study during 2021 addressed here occurred after the completion of the prior project in 2020. The past study (Hurd et al 2024, J Med Ent) focused on forest structural conditions that result from timber harvesting, microclimate, and the role of other hosts (i.e., deer), but did not address the role of small mammals. Therefore, we felt this gap warranted an additional study to examine how timber harvesting may affect small mammal communities, and, in turn, be able to speculate if these effects could have consequences for the differences in tick populations that we discovered in the prior study (Lines 100-102).

The authors include the species commonly know as the boreal red-backed vole in this ms. Note

that the scientific name has been officially changed back to Clethrionomys from Myodes in 2019. See Krystufek, B. (2019). "Back to the future: the proper name for red-backed voles is

Clethrionomys Tilesius and not Myodes Pallas". Mammalia. 84 (2): 214–217.

doi:10.1515/mammalia-2019-0067

>>>Thank you for catching this error! We have corrected the scientific name (Line 205).

Given that the manuscript is derived from data in August, are most of the data for ticks dealing with larvae? Or is there data on nymphs as well? If yes, they should be separated. If there is not data to examine nymphs, then the authors should make this clear in the text and may want to include ‘larval tick infestation’ in the title.

>>>Thank you for this feedback. Yes, all the tick data deals only with larvae as there was very limited detection of nymphs or adults. We agree it is necessary to make this clear and have clarified the results (Lines 209-210) and changed the title.

I would suggest a more detailed description of the type of harvests examined, and to consistently

refer to them directly in the ms. As an example, discussing forage trays ‘covered by understory

vegetation’ versus, presumably, high intensity harvest stands (lines 105-107)?

>>>Thank you for this comment. Within each harvest intensity treatment (low vs. high) we deployed foraging trays that were either covered or uncovered by understory vegetation, for a total of four distinct conditions: (1) low harvest/covered, (2) low harvest/uncovered, (3) high harvest/covered, and (4) high harvest/uncovered. We have added a description of this design to the methods section to improve clarity (Lines 169-174).

Broadly, this manuscript is about forest management techniques, but there is no information on the forest type or species composition, both of which may play some role in small mammal activity. For instance, the seasonal effect of seed production in a stand may be different if the producer is an American beech (Fragus grandifolia) versus a northern red oak (Quercus rubra). As this study was only conducted over a short period, for one month, the forest composition is a vital fact that has been left out. Similarly, where was this study conducted? Forest plots located on the southern coast may be very different from what are seen in northern industrial forests. Such factors may also influence the establishment and population size of Ix. scapularis, which seems to still be expanding its range in northern New England states. Based on the initial study cited in the text (Hurd et al. 2024), research occurred across, at least, three counties in Maine, some of which are geographically disparate.

>>>Thank you for this review, we agree additional study site description is needed. We have added additional information to the methods section about study location, forest type, and tree species composition (Lines 128-136).

Although this study focuses on foraging behavior, any information on nesting behavior, which may also play a role in co-feeding, as well as maintenance of the host-pathogen cycle?

>>>Thank you for this point, nesting behavior could also be an interesting component of this system. We did not gather any data on nesting behavior during this study, but agree it warrants consideration and have addressed it in the discussion section (Lines 273-279).

Finally, given that one of the key factors in limitation or mortality in Ix. scapularis is desiccation, is there micro-humidity data present for the study grids between the different forest harvests? It

would seem that this is an important question, if the investigators were looking at encounters

between hosts and questing ticks. Since the authors cites the use of dataloggers in the original

manuscript, cited above, why was that data not included here?

>>>Thank you for this comment, we have cited our microclimate data findings from our previous study (Lines 284-286).

Specifically, edits to be considered:

Line 57: Should the term reservoir host be used?

>>>We believe the term ‘host’ here is appropriate rather than ‘reservoir host’ because in this sentence we discuss small mammals as hosts for ectoparasites rather than as reservoir hosts for tickborne pathogens like Borrelia burgdorferi. The term reservoir host refers to the incubation and persistence of the pathogen, not the vector.

Methods section, line 67-95: It is stated later that camera traps were used, this should be included in this section, as should a description of the feeding trays. The authors do state later in the ms that the question of what types of animals visited the feeders are addressed by the use of cameras. They may want to include this earlier as it could be an important confounder if birds or larger mammals, such as raccoons were also visiting feeders.

>>>Thank you for this suggestion, we agree more detailed methods are necessary. We have added descriptions of the trays with trail cameras as well as the track plates (Lines 169-198). From the trail camera photos of the foraging trays, the only animals removing seed from the trays appeared to be small mammals like mice and possibly voles or shrews. There were two brief instances of raccoons inspecting the foraging trays, but the captured photos indicate no seed, or very limited seed, was removed during these times and the encounters were purely investigatory.

Results section, lines 114-122: What was the distance between sampling grids? Any issue with

migration from one area to another. That may address the uniformity of infection across the study.

>>>Thank you for this comment. We have added the distance between sampling grids (500 m) to the methods section (Line 130-131).

No data on the chipmunks???

>>>We chose to restrict our analyses to just Peromyscus leucopus data as chipmunk captures were so rare (only 10 captures study-wide, or less than 4% of total captures) and we only managed to gather full observations (i.e., tick burdens, ear tags, and ear biopsies) from 4 out of the 10 chipmunks. We felt the data was not substantial enough for robust analyses. We have added an explanation for the lack of chipmunk data to the results section (Lines 207-209).

Discussion section, lines 144-146: Maybe be specific about the species of squirrels discussed. Are you including chipmunks, which is a type of sciurid? Gray and red squirrels may differ greatly in their tick burdens as well, as one is more arboreal than the other.

>>>This is an important point, and we agree that specificity is required. We have included this detail to further clarify the findings from the studies we cited (Line 270).

---

## [Editor Report · Decision Letter 1]

9 Mar 2025

PONE-D-24-45269R1Short Communication: Timber harvesting impacts small mammal foraging behavior and larval tick infestationPLOS ONE

Dear Dr. Hurd,

Thank you for submitting your manuscript to PLOS ONE. After careful consideration, we feel that it has merit but does not fully meet PLOS ONE’s publication criteria as it currently stands. Therefore, we invite you to submit a revised version of the manuscript that addresses the points raised during the review process.

As indicated below, the response to the issue of deer activity was not sufficient.  The possible confounding influence of deer activity must be addressed in the discussion as a limitation of the study.

We look forward to receiving your revised manuscript.

Kind regards,

Sam R. Telford III

Academic Editor

PLOS ONE

Additional Editor Comments:

The response to the comment about deer activity is not adequate. There needs to be a specific discussion paragraph similar to what was published in Hurd et al 2024, which was referred to in the one line response to the review comment in lines 89-90 of the revised ms. In Hurd et al 2014, deer activity was not associated with trees/ha or basal area/ha, nor was the density of nymphs, but the discussion admitted that the experimental units were 1 ha and thus a small portion of deer home range. There was no specific estimator presented for deer activity for each of the 6 sites in that paper, just a MANOVA using specific measurements for each site. Did deer activity differ between the study sites, as measured in Hurd et al 2014, and is it not possible that the small mammal results are confounded by deer activity that could result in differential oviposition and thus density of available larvae?

---

## [Author Response · Author response to Decision Letter 2]

22 Apr 2025

Response to Editor Comments

As indicated below, the response to the issue of deer activity was not sufficient. The possible confounding influence of deer activity must be addressed in the discussion as a limitation of the study.

The response to the comment about deer activity is not adequate. There needs to be a specific discussion paragraph similar to what was published in Hurd et al 2024, which was referred to in the one line response to the review comment in lines 89-90 of the revised ms. In Hurd et al 2014, deer activity was not associated with trees/ha or basal area/ha, nor was the density of nymphs, but the discussion admitted that the experimental units were 1 ha and thus a small portion of deer home range. There was no specific estimator presented for deer activity for each of the 6 sites in that paper, just a MANOVA using specific measurements for each site. Did deer activity differ between the study sites, as measured in Hurd et al 2014, and is it not possible that the small mammal results are confounded by deer activity that could result in differential oviposition and thus density of available larvae?

>>>Thank you for this feedback, we have increased our discussion of deer activity. In the two years preceding this current study, deer activity did not significantly differ between the six experimental units used in this study. We added this clarification in the description of the study sites [Lines 69 – 72]. However, we agree the spatial limitation of the prior study may not have accurately captured deer activity, leaving the possibility of confounding our small mammal tick burden results. We have added a discussion of this limitation to the discussion section [Lines 199-207].

---

## [Editor Report · Decision Letter 2]

15 May 2025

Short Communication: Timber harvesting impacts small mammal foraging behavior and larval tick infestation

PONE-D-24-45269R2

Dear Dr. Hurd,

We’re pleased to inform you that your manuscript has been judged scientifically suitable for publication and will be formally accepted for publication once it meets all outstanding technical requirements.

Kind regards,

Sam R. Telford III

Academic Editor

PLOS ONE
---

## [Editor Report · Acceptance letter]

PONE-D-24-45269R2

PLOS ONE

Dear Dr. Hurd,

I'm pleased to inform you that your manuscript has been deemed suitable for publication in PLOS ONE. Congratulations! Your manuscript is now being handed over to our production team.

Kind regards,

on behalf of

Dr. Sam R. Telford III

Academic Editor

PLOS ONE